# A Computationally Efficient Case-Control Sampling Framework for G-Formula with Longitudinal Data

## Abstract

Estimating the causal effect of time-varying treatments on survival outcomes in large observational studies is computationally demanding, particularly when outcomes are rare. The iterative conditional expectation (ICE) estimator within the g-formula framework is effective but becomes computationally burdensome when bootstrapping is used for variance estimation. Additionally, the rarity of outcomes at each time point can create extreme class imbalance, leading to instability and convergence issues in logistic regression and related models. To address these challenges, we propose a novel case-control enhanced g-formula that integrates case-control sampling with ICE estimation. By strategically selecting informative subsets of data and applying appropriate reweighting, the method mitigates class imbalance, improves estimation stability, and substantially reduces computational cost, all while preserving consistency and asymptotic efficiency. We evaluate the method through simulations and validate it using a large-scale EHR cohort study on social and behavioral determinants of health (SBDH) and suicide risk, demonstrating its effectiveness for modeling rare outcomes in longitudinal data.

## 1 Introduction

Many observational studies in causal inference aim to estimate the causal effect of a time-varying treatment on a survival outcome. Examples include evaluating the impact of a long-term medication regimen on the survival of patients with chronic diseases (Hernán & Robins, 2006), assessing the effect of smoking cessation over time on lung cancer incidence (Kenfield et al., 2008), and studying the influence of lifestyle interventions on cardiovascular event-free survival (Knowler et al., 2002).

Under the standard causal assumptions of consistency, positivity, and exchangeability given measured confounders, the counterfactual probabilities of the time-to-event outcomes can be identified using the g-formula (Robins, 1986). Two widely-used forms of the g-formula are: (1) an expectation weighted by the joint distribution of covariates, treatments, and outcomes, and (2) an iterative conditional expectation over time. A straightforward approach to constructing estimators based on the g-formula is to first estimate each component and then plug these estimates into the g-formula. The estimator based on the first form of the g-formula is known as the *Non-Iterative Conditional Expectation (NICE)* estimator, which typically requires modeling the joint distribution of the confounders, treatments, and outcomes over time. The estimator based on the second representation is the *Iterative Conditional Expectation (ICE)* estimator, which relies on a sequence of conditional expectation models at each time point but avoids specifying the full joint distribution of confounders.

A major challenge in applying both the NICE and ICE estimators in large observational studies is the substantial computational burden, particularly when the outcome of interest is rare (King & Zeng, 2001). For example, in our analysis of a large-scale cohort study examining the effect of social determinants of health (SDOH) on suicide with around 130k individuals, fitting logistic regression models at each time step is computationally intensive. This difficulty arises primarily from the rarity of suicide and the large sample size, both of which contribute to slow convergence when estimating models for binary outcomes. In addition, because closed-form expressions for standard errors are difficult to derive in this setting, bootstrap methods are commonly used, further increasing the computational burden through repeated resampling and model fitting. This issue

is particularly acute for g-formula methods, where models must be fit repeatedly across multiple time points, compounding the overall cost. These considerations highlight the critical importance of developing computationally efficient methods. To address this challenge, we introduce a case-control design, an established and efficient approach to studying rare outcomes due to its reduced sample size requirements and cost-effectiveness (Breslow, 1996; van der Laan, 2008). Although the outcome is rare relative to controls at each time point, the large overall sample size ensures that a sufficient number of cases are observed for reliable estimation. In this setting, case-control sampling offers a promising solution. Traditionally, case-control studies involve sampling a fixed number of cases (individuals with the outcome) and a matched set of controls (individuals without the outcome), enabling efficient estimation of parameters such as odds ratios while substantially reducing computational cost. Building on this idea, we propose a novel integration of the case-control design with the g-formula to handle time-varying treatments and survival outcomes. By leveraging the sampling efficiency of the case-control approach, our method significantly reduces computation time while maintaining estimator consistency. We demonstrate the effectiveness of the proposed method through extensive simulation studies and a real data application involving suicide as a rare outcome.

## 2 RELATED WORK

A range of methodological frameworks have been developed to estimate causal effects in longitudinal settings where treatments, covariates, and outcomes evolve over time. These include the g-formula, marginal structural models (MSMs), and structural nested models (SNMs), which are designed to address challenges such as time-varying confounding and to evaluate causal effects under history-dependent treatment strategies (Hernán & Robins, 2020; Wen et al., 2021; Robins et al., 2000; Vansteelandt & Joffe, 2014). All three approaches aim to correct for time-dependent confounding that arises when past exposures influence subsequent covariates, but they differ in how they achieve this goal. MSMs rely on inverse-probability weighting to construct a pseudo-population in which treatment assignment is independent of past confounders. These models are relatively straightforward to implement using standard regression tools and yield marginal causal effects. However, their performance can be compromised by highly variable or extreme weights, especially in settings with limited overlap, leading to unstable or biased estimates. SNMs estimate causal effects via g-estimation of blip functions, which quantify the change in the counterfactual outcome induced by deviating from a reference treatment path at each time point, conditional on the observed history. With correctly specified treatment and outcome models, g-estimation attains the semiparametric efficiency bound and yields history-specific causal contrasts. In practice, however, SNMs rely on computationally demanding, sometimes unstable iterative equations, are highly sensitive to misspecification of either model, and yield blip parameters that have limited direct clinical interpretability.

The parametric g-formula models each component of the data-generating process and can be implemented through two algebraically equivalent forms which will be discussed in Section 3. Although it requires correct specification of all sub-models, the g-formula avoids the weight instability inherent to MSMs and the iterative g-estimation burden of SNMs, accommodates both continuous and time-to-event outcomes, and provides clear population-level contrasts for complex static or dynamic treatment strategies. Because of this finite-sample stability, modelling flexibility, and transparent counterfactual interpretation, we use the g-formula to examine how evolving social determinants of health influence suicide-related outcomes over time.

In addition to these well-established frameworks, subsampling has emerged as an important strategy for addressing the computational challenges inherent in large-scale data. Rather than analyzing the entire cohort, one can draw informative subsets of the data to reduce computational burden while still obtaining valid inference. In the rare-event settings, it has been shown that retaining all cases while subsampling a sufficiently large number of controls, with appropriate reweighting yields estimators that are consistent and, under certain asymptotic regimes, have the same asymptotic distribution as the full-sample maximum likelihood estimator (Wang, 2020). This reflects that when outcomes are highly imbalanced, most of the information comes from the rare cases, so additional controls contribute little beyond a certain point. Consequently, subsampling provides a principled way to extend g-formula–based methods and related estimators to massive datasets where full-sample estimation is computationally prohibitive.

## 3 METHODOLOGY

**Study Design**  We consider a longitudinal study with discrete time points $j = 0, \ldots, T$, where individuals are followed over time. At each time $j$, covariates $L_j$ and treatment $A_j$ are observed, followed by the censoring indicator $C_{j+1}$ and event indicator $Y_{j+1}$. Censoring ($C_{j+1} = 1$) indicates that the individual is no longer observed from time $j + 1$ onward. This may occur in practice due to loss to follow-up, withdrawal, or administrative censoring at the end of the study period. The survival outcome is represented such that if an individual dies at time $k$, then $Y_k = 1$ and $Y_j = 0$ for all $j < k$. The data are temporally ordered as $(L_j, A_j, C_{j+1}, Y_{j+1})$, with $Y_0 = C_0 = 0$ and $\bar{L}_{-1} = \bar{A}_{-1} = \emptyset$ by convention. As an illustration, an individual who dies at time 2 would have the observed data sequence $(L_{-1} = \emptyset, A_{-1} = \emptyset, C_0 = 0, Y_0 = 0, L_0 = l_0, A_0 = a_0, C_1 = 0, Y_1 = 0, L_1 = l_1, A_1 = a_1, C_2 = 0, Y_2 = 1)$.

We use $Y_t^g$ to denote the potential outcome at time $t$ had an individual followed a deterministic treatment strategy $g$. A deterministic strategy $g$ specifies the treatment $A_j$ to be assigned at each time $j$ based on the observed history up to that point, $(\bar{L}_j, \bar{A}_{j-1})$. Our goal is to identify and estimate $E(Y_t^g)$, the mortality risk by time $t$ if all individuals in the population were to follow strategy $g$.

**G-formula**  Identification of $E(Y_t^g)$ using the g-formula (Robins, 1986) relies on three key assumptions:

1. *Consistency:* If an individual's observed treatment history matches the strategy $g$ up to time $j$, i.e., $\bar{A}_j = \bar{A}_j^g$, then their observed covariates and outcomes also equal the counterfactuals: $\bar{L}_j = \bar{L}_j^g$ and $\bar{Y}_{j+1} = \bar{Y}_{j+1}^g$.

2. *Exchangeability:* At each time $j$, future counterfactual outcomes are independent of treatment and censoring decisions, conditional on past covariate and treatment history and being uncensored and event-free:

$$(Y_{j+1}^g, \ldots, Y_J^g) \perp (A_j, C_{j+1}) \mid \bar{L}_j = \bar{l}_j, \bar{A}_{j-1} = \bar{a}_{j-1}^g, C_j = Y_j = 0.$$

3. *Positivity:* If a covariate and treatment history occurs with positive probability under the observed data, then the probability of receiving the treatment assigned by strategy $g$ and remaining uncensored at the next time must also be positive:

$$f_{\bar{L}_j, \bar{A}_{j-1}, C_j, Y_j}(\bar{l}_j, \bar{a}_{j-1}^g, 0, 0) > 0 \;\Rightarrow\; f_{A_j, C_{j+1} \mid \bar{L}_j, \bar{A}_{j-1}, C_j, Y_j}(a_j^g, 0 \mid \bar{l}_j, \bar{a}_{j-1}^g, 0, 0) > 0.$$

Under these assumptions, the counterfactual risk $E(Y_T^g)$ can be identified via the g-formula:

$$E(Y_T^g) = \sum_{\bar{l}_{T-1}} \sum_{t=1}^{T} P(Y_t = 1 \mid Y_{t-1} = C_t = 0, \bar{L}_{t-1} = \bar{l}_{t-1}, \bar{A}_{t-1} = \bar{a}_{t-1}^g) \qquad (1)$$

$$\prod_{s=0}^{t-1} P(Y_s = 0 \mid Y_{s-1} = C_s = 0, \bar{L}_{s-1} = \bar{l}_{s-1}, \bar{A}_{s-1} = \bar{a}_{s-1}^g) f(l_s \mid Y_s = C_s = 0, \bar{L}_{s-1} = \bar{l}_{s-1}, \bar{A}_{s-1} = \bar{a}_{s-1}^g).$$

Equation (1) is known as the non-iterative conditional expectations (NICE) representation. Alternatively, using iterated conditional expectations (ICE), $E(Y_T^g)$ can be identified as:

$$E(Y_T^g) = E_{f_{L_0}} \Big( E_{f_{Y_1}} \Big[ Y_1 + \cdots E_{f_{Y_{T-1}}} \big\{ Y_{T-1}(1 - Y_{T-2}) + E_{f_{L_{T-1}}} \big[ E_{f_{Y_T}} \{ Y_T(1 - Y_{T-1}) \mid$$

$$\bar{Y}_{T-1}, C_T = 0, \bar{L}_{T-1}, \bar{A}_{T-1} = \bar{A}_{T-1}^g \} \mid \bar{Y}_{T-1}, C_{T-1} = 0, \bar{L}_{T-2}, \bar{A}_{T-2} = \bar{A}_{T-2}^g \big] \mid \quad (2)$$

$$\bar{Y}_{T-2}, C_{T-1} = 0, \bar{L}_{T-2}, \bar{A}_{T-2} = \bar{A}_{T-2}^g \} \ldots \mid C_1 = 0, A_0 = A_0^g, L_0 \Big] \Big).$$

In this paper, we focus on the ICE formula (2). However, our methods are also applicable to the NICE estimator, as demonstrated in our simulation studies. Based on (2), an algorithm for estimating $E(Y_T^g)$ is presented in Algorithm A1 in Appendix A2. If the parametric models in lines 1 and 6 in Algorithm A1 are correctly specified, this estimator will be consistent.

**An Illustrative Example**  In Algorithm A1, we estimate a model at each time point using all available samples. In large cohorts with rare outcomes, the excess of controls makes computation

intensive, especially with multiple time points and bootstrapping. To improve efficiency, we propose to apply Algorithm A1 to a case-control sample that retains all cases while subsampling controls at each time point, reducing computational burden without sacrificing estimation reliability.

For ease of illustration, we consider two time points, $t = 0, 1$, in the absence of censoring. Suppose the total number of individuals is $N$, among whom $C$ experience the event. Let $c_t$ denote the number of cases who die at time $t$, so that $C = c_0 + c_1$. At each time $t$, we include all cases who die at $t$ and randomly sample $Jc_t$ controls with replacement from individuals alive at $t$, where $J$ is the sampling ratio. Thus, the total case-control sample size is $(J+1)(c_0 + c_1)$. If an individual is sampled at both time points, they are treated as separate observations.

Suppose we aim to estimate the expectation of a covariate $X$. The most straightforward estimator from the full sample would be $\hat{E}(X) = N^{-1} \sum_{i=1}^{N} X_i$. Now we would like to draw a subsample and use the subsample to obtain a valid estimate. The original sample is divided into three non-overlapping groups: $G_0 = \{i : Y_0 = 1\}$ (those who die at $t = 0$); $G_1 = \{i : Y_1 = 1, Y_0 = 0\}$ (those who die at $t = 1$); and $G_2 = \{i : Y_1 = 0\}$ (those alive at $t = 1$). Table 1 gives a description of the subsample. Here, the column "weight" gives the weight associated with each category, the column "group" lists the groups included in the category, and the column "count" indicates the number of times a person in this category is counted. Using time 0 control as an example, the probability that a person alive at time 0 is selected into the subsample is $(Jc_0)/(N - c_0)$. Therefore, the count is the product of the weight and this selection probability.

Table 1: Case-control sample breakdown with two time points and no censoring

|  | size | weight | group | count |
|---|---|---|---|---|
| time 0 case | $c_0$ | $w_0$ | $G_0$ | $w_0 \cdot 1$ |
| time 0 control | $Jc_0$ | $m_0$ | $G_1, G_2$ | $m_0 \cdot \frac{Jc_0}{N - c_0}$ |
| time 1 case | $c_1$ | $w_1$ | $G_1$ | $w_1 \cdot 1$ |
| time 1 control | $Jc_1$ | $m_1$ | $G_2$ | $m_1 \cdot \frac{Jc_1}{N - c_0 - c_1}$ |

An individual in $G_0$ is always included and thus contributes $w_0 \cdot 1$. An individual in $G_1$ may appear as a time 0 control with probability $Jc_0/(N - c_0)$ and as a time 1 case with probability 1, so its expected contribution is $w_1 \cdot 1 + m_0 \cdot Jc_0/(N - c_0)$. An individual in $G_2$ may appear as a time 0 control with probability $Jc_0/(N - c_0)$ and as a time 1 control with probability $Jc_1/(N - c_0 - c_1)$, giving an expected contribution of $m_1 \cdot J_{c1}/(N - c_0 - c_1) + m_0 \cdot Jc_0/(N - c_0)$. In the full sample, each individual is counted exactly once. To ensure that the case-control subsample is representative of the full sample, we require that the expected contribution per individual is equal across groups. This balancing condition leads to

$$w_0 = w_1 + m_0 \frac{Jc_0}{N - c_0} = m_1 \frac{Jc_1}{N - c_0 - c_1} + m_0 \frac{Jc_0}{N - c_0},$$

which reduces to

$$w_1 = m_1 \frac{Jc_1}{N - c_0 - c_1}, \quad w_0 - w_1 = m_0 \frac{Jc_0}{N - c_0}.$$

Since there are 2 equations with 4 parameters $(w_0, w_1, m_0, m_1)$, we have 2 degrees of freedom. If we let $w_0 = 1/N$, then all the weights add up to 1: $c_0 w_0 + c_1 w_1 + Jc_1 m_1 + Jc_0 m_0 = 1$. There is one degree of freedom remaining: if we further let $w_1 = 1/N$, then $m_0 = 0, m_1 = (N - c_0 - c_1)/(JNc_1)$. It is also possible to set $w_1$ to other values and obtain corresponding values of $m_0$ and $m_1$. Additional examples of weight combinations are provided in the Appendix A3.

When we estimate a model, such as fitting a logistic regression, we are actually solving model parameters from estimating equations. In many cases, the estimating equation is from the maximum likelihood estimation process by setting the gradient to zero. Now we verify that the above weighting approach also applies to estimating equations. We denote $O^*$ as the whole population, and $O = (O_{0c}, O_{0m}, O_{1c}, O_{1m})$ as the case-control sample. Suppose we have $E_0^*\{S_0(O^*)\} = 0$, where $E_0^*$ means the expectation is taken with respect to the true distribution of the whole population, and $S_0$ is the original estimating equation. We let $S(O) = c_1 w_1 S_0(O_{1c}) + Jc_1 m_1 S_0(O_{1m}) + c_0 w_0 S_0(O_{0c}) +$

$Jc_0 m_0 S_0(O_{0m})$, be the weighted estimating equation for the case-control sample. Then it actually holds that

$$E\{S(O)\} = w_0 N E_0^*\{S_0(O^*)\} = 0, \tag{3}$$

where the expectation $E$ is taken with respect to the sampling distribution induced by the case-control sampling design. The proof of (3) is relegated to the Appendix A4. Therefore, with appropriate weights, $E_0^*\{S_0(O^*)\} = 0$ if and only if $E\{S(O)\} = 0$, implying that the model parameters can be estimated by solving $\hat{E}\{S(O)\} = 0$ using the case-control sample.

**General Results with Censoring** Now we extend our setting to include $k$ time points, $t = 0, 1, ..., k - 1$, as well as censoring. The number of censored individuals at time $t$ is denoted by $s_t$. We also allow the sampling ratios $J_t$, $K_t$ to vary across time points. At each time point $t$, we sample $K_t c_t$ from people who are censored at time $t$ except for the last time point, and sample $J_t c_t$ controls from people who are alive at time $t$. Then the case-control sample has a size of $\sum_{t=0}^{k-1} c_t + \sum_{t=0}^{k-1} J_t c_t + \sum_{t=0}^{k-2} K_t c_t$. We do not sample censored individuals at the last time point because their $Y_{k-1}$ is unobserved, preventing their use in fitting logistic or other classification models in the first step of Algorithm A1. However, at earlier time points, censored individuals can contribute to estimating their counterfactual risk, so they may be included.

The population is partitioned into $2k + 1$ groups: $G_0, G_1, \ldots, G_{2k-1}, G_{2k}$, where $G_{2i}$ is the group of people who are censored at time $i$ for $i = 0, \ldots, k - 1$, $G_{2i+1}$ is the group of people who die at time $i$ for $i = 0, \ldots, k - 1$, and $G_{2k}$ is the group of people who are alive at time $k - 1$. Table A1 in Appendix A1 gives a description of the subsample with $k$ time points, with the "count" column constructed in the same manner as in Table 1. We use $w_t$ to denote the weights for cases, $m_t$ for sampled controls, and $\ell_t$ for sampled censored individuals at time $t$.

Requiring that each individual in groups $G_0$ through $G_{2k}$ be counted the same number of times (detailed in Appendix A1) leads to the following conditions:

$$w_t = \ell_t \frac{K_t c_t}{s_t}, \quad t = 0, \ldots, k - 1,$$

$$w_{k-1} = m_{k-1} \frac{J_{k-1} c_{k-1}}{N - \sum_{j=0}^{k-1} c_j - \sum_{j=0}^{k-1} s_j}, \tag{4}$$

$$w_t - w_{t+1} = m_t \frac{J_t c_t}{N - \sum_{j=0}^{t} c_j - \sum_{j=0}^{t} s_j}, \quad t = 0, \ldots, k - 2.$$

Since there are $2k$ equations with $3k$ parameters $(w_0, \ldots, w_{k-1}, m_0, \ldots, m_{k-1}, \ell_0, \ldots, \ell_{k-1})$, we have $k$ degrees of freedom. If we let $w_0 = 1/N$, then all the weights add up to 1:

$$\sum_{t=0}^{k-1} c_t w_t + \sum_{t=0}^{k-1} J_t c_t m_t + \sum_{t=0}^{k-1} K_t c_t \ell_t = w_0 N = 1.$$

Letting $K_t c_t = s_t$ for $t = 0, \ldots, k - 1$, we have $\ell_t = w_t$. This implies that, at each time point $t$, if all censored individuals are included, they should be assigned the same weights as the cases at time $t$. We adopt this design in our paper. Under this setup, we have $k$ equations for $2k$ unknowns, resulting in infinitely many possible solutions. In this paper, we consider one specific set of weights: $\ell_0 = w_0 = \frac{c_0}{N - s_0}$, $m_0 = \frac{1}{J} \frac{N - c_0 - s_0}{N - s_0}$, $\ell_t = w_t = m_t = 0$, for $t = 1, \ldots, k - 1$. It is straightforward to verify that this set of weights satisfies (4). This choice corresponds to a simple weighting scheme that includes only individuals from time $t = 0$. Note that in Algorithm A1, which applies to the complete data, the current time point $k + 1$ can be treated as $t = 0$ when fitting the models in line 6. Additionally, because a case at a later time point may be sampled as a control at an earlier time point, an individual may receive different weights across time. The corresponding ICE estimator based on the constructed case-control sample is presented in Algorithm 1.

**Theorem 1.** *The estimator from Algorithm 1 is consistent for $E(Y_T^g)$ under the same assumptions that ensure the consistency of the ICE estimator in Algorithm A1.*

The proof of Theorem 1 is in Appendix A5.

---

**Algorithm 1** Case-Control Sampling and Estimation Procedure with Censoring

---

1: Generate a case-control sample:
2: **for** each time point $t$ **do**
3:     Keep all $c_t$ cases at time $t$ (i.e., those who die at time $t$).
4:     **if** $t < k - 1$ **then**
5:         Randomly draw $J_t c_t$ controls from those alive at time $t$ with replacement.
6:         Assign weight $w_t$ to each case at time $t$.
7:         Assign weight $w_t$ to each censored individuals at time $t$.
8:         Assign weight $m_t$ to controls sampled at time $t$.
9:     **else if** $t = k - 1$ **then**
10:        Randomly draw $J_{k-1} c_{k-1}$ controls from those alive (excluding censored individuals) at time $k - 1$ with replacement.
11:        Assign weight $w_{k-1}$ to each case at time $k - 1$.
12:        Assign weight $m_{k-1}$ to controls sampled at time $k - 1$.
13:     **end if**
14: **end for**
15: Fit a weighted regression of $Y_T$ on $\bar{L}_{T-1}$ and $\bar{A}_{T-1}$ among individuals with $Y_{T-1} = C_T = 0$ using the associated weights, and estimate parameter $\theta_{T,T-1}$.
16: Obtain predicted values $\hat{h}^g_{T,T-1}$ from

$$E(Y_T \mid Y_{T-1} = C_T = 0, \bar{L}_{T-1}, \bar{A}_{T-1} = \bar{A}^g_{T-1}; \hat{\theta}_{T,T-1})$$

    by fixing $\bar{A}_{T-1} = \bar{A}^g_{T-1}$ among individuals with $Y_{T-1} = C_{T-1} = 0$. Set $q = 2$.
17: **while** $q \leq T$ **do**
18:     Let $k = T - q$.
19:     Define

$$\hat{Q}^g_{T,k+1} = \begin{cases} \hat{h}^g_{T,k+1} & \text{if } Y_{k+1} = 0, \\ 1 & \text{if } Y_{k+1} = 1. \end{cases}$$

20:     Fit a weighted regression of $\hat{Q}^g_{T,k+1}$ on $\bar{L}_k$ and $\bar{A}_k$ among individuals with $Y_k = C_{k+1} = 0$ using the associated weights, and estimate parameter $\theta_{T,k}$.
21:     Obtain predicted values $\hat{h}^g_{T,k}$ from

$$E(\hat{Q}^g_{T,k+1} \mid Y_k = C_{k+1} = 0, \bar{L}_k, \bar{A}_k = \bar{A}^g_k; \hat{\theta}_{T,k})$$

    by fixing $\bar{A}_k = \bar{A}^g_k$ among individuals with $Y_k = C_k = 0$.
22:     Set $q = q + 1$.
23: **end while**
24: Compute the weighted average of $\hat{h}^g_{T,0}$ using the generated weights to estimate $E(Y^g_T)$.

---

## 4 NUMERICAL EXAMPLES

### 4.1 SIMULATED LONGITUDINAL DATA

We conducted simulation studies to compare the computational cost and estimation efficiency of the proposed case-control estimator with its complete-data counterpart. The sampling ratios were set at $1 : 5$, $1 : 10$, and $1 : 20$. Each dataset contained $N = 30,000$ individuals followed over six time points. Baseline covariates $L_b$ included four categorical variables, and at each time $t$, the data consisted of ten binary time-varying covariates $L_t \in \mathbb{R}^{10}$, a binary treatment indicator $A_t$, a binary censoring indicator $C_t$, and a death indicator $Y_t$. The observed data for each individual were $(L_b, L_0, A_0, C_1, Y_1, \ldots, L_5, A_5, C_6, Y_6)$. A first-order (lag-1) dependence structure was assumed for covariates, censoring, and outcomes, and all variables were generated at the individual level using logistic regression models, following the design of Wen et al. (2021).

The details of data generation are as follows. Baseline covariates $L_b$ was generated from multinomial distributions with the probability vectors $(1, 2)$, $(1, 1, 1, 1, 1, 1)$, $(5, 2, 1, 1, 4)$, and $(2, 3, 1, 4)$,

respectively; $L_0 \sim \text{Ber}(0.5)$, and $A_0 \sim \text{Ber}(\text{expit}(L_0^\top \eta))$. Moreover, for $t \geq 1$,

$$C_t \sim \text{Ber}(\text{expit}(-6 + A_{t-1} + \gamma^\top L_{t-1})), \qquad Y_t \sim \text{Ber}(\text{expit}(-6 + 2A_{t-1} + \beta^\top L_{t-1})),$$

$$L_t \sim \text{Ber}(\text{expit}(\eta A_{t-1} + ML_{t-1})), \qquad A_t \sim \text{Ber}(\text{expit}(A_{t-1} + \alpha^\top L_t)),$$

where the entries of the coefficient vectors $\alpha$, $\beta$, $\gamma$, $\eta$, and the matrix $M$ were independently generated from a standard normal distribution and then fixed throughout the simulations. This setting yields a prevalence rate of about 1%; results for additional scenarios with higher prevalence rates are provided in Appendix A6. We were interested in estimating the counterfactual risks under two deterministic treatment strategies: always treated and never treated, i.e., $\bar{A}_5 = (1, 1, 1, 1, 1, 1)$ and $\bar{A}_5 = (0, 0, 0, 0, 0, 0)$.

We simulated 100 independent datasets under the same parameter configuration. For each dataset, we performed patient-level bootstrap resampling with $B = 100$ replicates and applied both Complete and Case-control versions of the NICE and ICE estimators with logistic regression. We also explored more computationally intensive machine learning approaches with superlearner, an ensemble framework that combines multiple candidate algorithms to optimize predictive performance. For each estimator, we computed the bootstrap mean and standard deviation of the estimated target quantity. For the ICE method with logistic regression, we followed the R package `gfoRmula` (McGrath et al., 2020) and implemented our case-control version. Superlearner models were implemented using the R package `SuperLearner` (Polley et al., 2019) with base learners supporting weighted samples and probabilistic outcomes, including logistic regression and XGBoost.

**Computation Time**   Computation times for logistic regression and superlearner were summarized in Table 2 and Table A2. Each entry is the average of 10,000 analyses, that is, 100 bootstrap resamples applied to each of 100 independently simulated data sets. For the case-control version we report both (i) the total runtime, which includes construction of the case-control sample plus model fitting, and (ii) the model-fitting time alone, allowing the additional cost of the sampling step to be isolated. Each bootstrap resample and model fitting was conducted on a Linux HPC using a single node, equipped with either an Intel Skylake or AMD Epyc processor (Epyc64 or Epyc128 architecture). Each task was allocated 8 CPU cores and 4 GB of memory. The analyses were performed using R version 4.4.1, loaded via the module environment.

Table 2: Summary of the computation time (in secs) with logistic regression

|  | NICE ($\bar{A} = 0$) | ICE ($\bar{A} = \bar{0}$) | NICE ($\bar{A} = \bar{1}$) | ICE ($\bar{A} = \bar{1}$) |
|---|---|---|---|---|
| Complete | 11.79 (2.27) | 4.56 (0.95) | 11.83 (2.32) | 4.26 (0.84) |
| (*model-fitting time*) |  |  |  |  |
| Case-control ($J = 20$) | 6.61 (1.90) | 1.78 (0.57) | 6.64 (2.00) | 1.72 (0.73) |
| Case-control ($J = 10$) | 5.83 (1.74) | 1.17 (0.60) | 5.83 (2.17) | 1.14 (0.73) |
| Case-control ($J = 5$) | 5.89 (1.34) | 0.83 (0.49) | 5.90 (1.54) | 0.80 (0.54) |
| (*total runtime*) |  |  |  |  |
| Case-control ($J = 20$) | 8.09 (2.66) | 2.60 (1.20) | 8.11 (2.76) | 2.55 (1.46) |
| Case-control ($J = 10$) | 6.91 (2.34) | 1.94 (1.22) | 6.94 (2.99) | 1.94 (1.47) |
| Case-control ($J = 5$) | 6.77 (1.75) | 1.54 (1.01) | 6.79 (1.95) | 1.50 (1.01) |

Here the computation time (Table 2 and Table A2) referred to the average runtime for a single subsample run. However, because we used the bootstrap to estimate standard errors, the total computation time for one Monte Carlo run scaled linearly with the number of bootstrap replications. As expected, the overall computation time increased considerably when using superlearner compared to the logistic regression results, due to the added complexity of ensemble learning. Our case-control approach achieved a dramatic reduction in computation time relative to the complete-data implementation. These results highlighted the practical efficiency and scalability of our method, particularly in scenarios involving complex models or large-scale datasets where full-sample estimation becomes computationally demanding.

**Risk Estimates**   Table 3 presents the estimates of the ICE and NICE estimators along with their bootstrap standard errors. The results indicate that increasing $J$ from 5 to 20 reduces the standard

Table 3: Summary of the average of the ICE and NICE estimates based on bootstrap mean (multiplied by 100), with average of bootstrap SE in parentheses

| | | ICE | | | | NICE | | |
|---|---|---|---|---|---|---|---|---|
| | | Case-control | | Complete | | Case-control | | Complete |
| Time | $(J = 5)$ | $(J = 10)$ | $(J = 20)$ | | $(J = 5)$ | $(J = 10)$ | $(J = 20)$ | |
| | | | | $\bar{A} = \bar{1}$ | | | | |
| 1 | 2.15 (0.19) | 2.11 (0.17) | 2.09 (0.15) | 2.08 (0.14) | 2.22 (0.10) | 2.21 (0.09) | 2.21 (0.08) | 2.15 (0.08) |
| 2 | 3.97 (0.21) | 3.94 (0.19) | 3.93 (0.18) | 3.91 (0.17) | 3.69 (0.13) | 3.68 (0.12) | 3.67 (0.12) | 3.57 (0.11) |
| 3 | 5.44 (0.22) | 5.40 (0.21) | 5.38 (0.20) | 5.36 (0.19) | 4.96 (0.16) | 4.95 (0.15) | 4.94 (0.15) | 4.80 (0.14) |
| 4 | 6.55 (0.24) | 6.49 (0.22) | 6.46 (0.21) | 6.43 (0.20) | 6.17 (0.20) | 6.16 (0.19) | 6.15 (0.18) | 5.97 (0.17) |
| 5 | 7.69 (0.26) | 7.62 (0.24) | 7.59 (0.23) | 7.55 (0.22) | 7.39 (0.23) | 7.38 (0.22) | 7.37 (0.22) | 7.15 (0.20) |
| 6 | 8.82 (0.27) | 8.74 (0.25) | 8.70 (0.24) | 8.66 (0.23) | 8.61 (0.27) | 8.59 (0.26) | 8.58 (0.25) | 8.33 (0.23) |
| | | | | $\bar{A} = \bar{0}$ | | | | |
| 1 | 0.29 (0.05) | 0.29 (0.05) | 0.30 (0.05) | 0.30 (0.04) | 0.31 (0.03) | 0.31 (0.03) | 0.31 (0.03) | 0.30 (0.03) |
| 2 | 0.58 (0.08) | 0.59 (0.08) | 0.59 (0.07) | 0.59 (0.07) | 0.56 (0.05) | 0.56 (0.05) | 0.56 (0.05) | 0.55 (0.05) |
| 3 | 0.87 (0.11) | 0.87 (0.10) | 0.87 (0.10) | 0.87 (0.10) | 0.80 (0.07) | 0.80 (0.07) | 0.80 (0.07) | 0.78 (0.07) |
| 4 | 1.13 (0.14) | 1.13 (0.13) | 1.13 (0.12) | 1.13 (0.12) | 1.02 (0.09) | 1.02 (0.09) | 1.02 (0.09) | 0.99 (0.09) |
| 5 | 1.47 (0.17) | 1.46 (0.16) | 1.46 (0.15) | 1.45 (0.14) | 1.24 (0.11) | 1.24 (0.11) | 1.24 (0.11) | 1.20 (0.10) |
| 6 | 1.82 (0.19) | 1.80 (0.18) | 1.79 (0.17) | 1.78 (0.16) | 1.45 (0.13) | 1.45 (0.13) | 1.45 (0.13) | 1.41 (0.12) |

errors, bringing them closer to those from the complete-data scenario. The boxplots for the bootstrap mean across the 100 simulated datasets were given in Figure 1. For both the ICE and NICE estimators, the case-control versions yield results comparable to their complete-data counterparts.

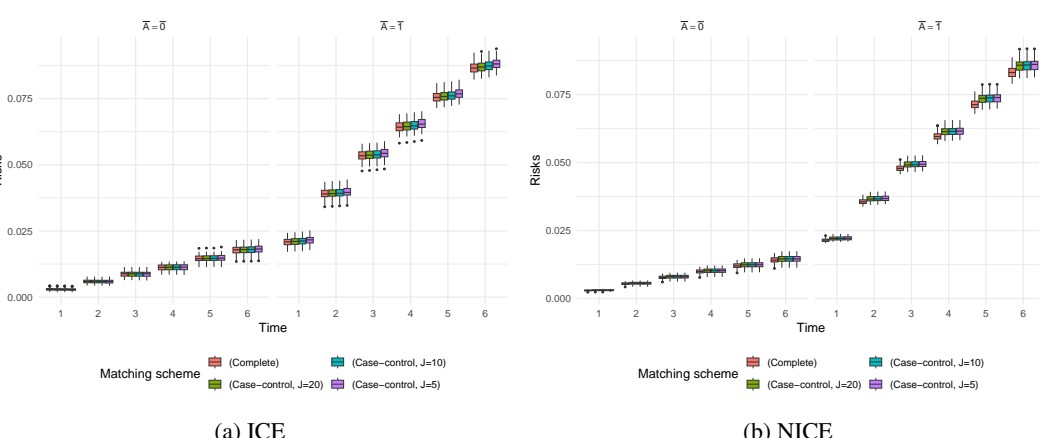

(a) ICE  (b) NICE

Figure 1: Risk estimates for always or never exposed to treatment.

## 4.2 REAL-WORLD LONGITUDINAL DATA

We applied the proposed case-control ICE estimators to a real-world cohort of 127,399 U.S. veterans who were discharged from short-term psychiatric hospital admissions at Veterans Health Administration (VHA) hospitals between January 1, 2016, and December 31, 2017. We followed up these patients until the end of December 31, 2018 or the event of interest, whichever comes first. Here we choose death by suicide (440 Veterans [0.35%]) as the event of interest and presence of any social or behavioral determinants of health (SBDH) (Bompelli et al., 2021) as treatment $A$. Time-fixed baseline covariates ($L_b$) include race, gender, age groups[1] and marital status (Table A6 in Appendix) while time-varying covariates ($L_t$) include 17 clinical comorbidities and 7 mental health disorders. The analysis was conducted over five consecutive 6-month intervals. Following the work by Mitra et al. (2023), we extracted SBDH information from both structured data (International Classification of Diseases [ICD] codes, stop codes) and unstructured data (clinical notes, using a fine-tuned transformer-based (Vaswani et al., 2017) deep-learning model). More about SBDH, clinical comorbidites and mental health disorders are available in Appendix A7.

---

[1]Although age changes over time, this is deterministic and hence we consider this as a time-fixed covariate.

Given the low incidence rate of suicide in our cohort (718 cases out of $125,928$ patients across 5 time points, as detailed in Table A5 in Appendix), we adopted an adaptive sampling ratio across different time points. For each case at time point from 1 to 5, 30, 30, 50, 80, and 80 control patients were sampled, respectively. Table 4 reports computation times, showing that the case-control ICE estimator runs in roughly one-quarter of the time needed for the complete-data version. Table 5 shows the estimated risks. The point estimates are similar for both methods, but the case-control approach has a larger standard error at the last time point.

Table 4: Application: Computation time of ICE method with complete or case-control data (in secs)

|  | $\bar{A} = \bar{0}$ | $\bar{A} = \bar{1}$ |
|---|---|---|
| Complete | 24.66 (11.98) | 21.25 (13.35) |
| Case-control (*total runtime*) | 5.73 (1.77) | 5.70 (1.83) |
| Case-control (*model-fitting time*) | 3.64 (1.63) | 3.58 (1.77) |

Table 5: Application: Risk estimates from 100 bootstrap samples (standard deviations in parentheses); all values multiplied by 100

| Time | $\bar{A} = \bar{0}$ | | $\bar{A} = \bar{1}$ | |
|---|---|---|---|---|
|  | Complete | Case-control | Complete | Case-control |
| 1 | 0.12 (0.02) | 0.12 (0.02) | 0.07 (0.01) | 0.07 (0.01) |
| 2 | 0.19 (0.02) | 0.20 (0.03) | 0.14 (0.01) | 0.15 (0.02) |
| 3 | 0.24 (0.03) | 0.25 (0.03) | 0.23 (0.02) | 0.24 (0.02) |
| 4 | 0.27 (0.03) | 0.28 (0.03) | 0.31 (0.02) | 0.33 (0.02) |
| 5 | 0.29 (0.03) | 0.31 (0.05) | 0.36 (0.02) | 0.40 (0.06) |

## 5 DISCUSSION

In this paper, we introduced a case-control sampling framework to improve the computational efficiency of estimating counterfactual risks under time-varying treatment regimes. Our numerical experiments demonstrate that the proposed approach substantially reduces computation time relative to complete-data implementations, while maintaining comparable estimation accuracy. In practice, bootstrap resampling is widely used to estimate standard errors, often with $B = 1000$ or more replications. Even a modest per-run speedup can translate into hours of savings across all replications. This advantage becomes even more significant in online or routinely updated analyses, where models must be refitted repeatedly as new data arrive. Our framework therefore provides a scalable alternative without sacrificing consistency. The method is also highly flexible: the construction of weights is not unique, and different weighting schemes can be tailored to specific analytic goals. For instance, one may choose whether to include censored individuals, or whether to sample cases and controls from future time points. This flexibility raises a natural question for future research: which weighting strategies yield the most efficient estimators in terms of variance when the number of sampled controls is comparable to the number of cases? Prior work on optimal sampling strategies in logistic regression models (e.g., Fithian & Hastie, 2014; Wang et al., 2021) provides useful theoretical guidance.

In practice, the iterative fitting procedure can cause risk-estimate bias to accumulate over time, particularly in later periods. The main culprit is the difficulty of fitting logistic (or similar) models when outcomes are rare, which leads to estimation instability. A pragmatic fix is to use penalized methods, for example, Firth's logistic regression, at each step to mitigate the bias. More flexible machine learning models could also be integrated into the ICE framework to boost predictive accuracy and robustness in high-dimensional or otherwise complex settings. Beyond ICE, the proposed weighting scheme can also be naturally incorporated into doubly robust estimators, such as longitudinal TMLE (van der Laan & Gruber, 2012) or AIPTW, by applying the weights to the influence-function contributions. Exploring these extensions represents an important direction for future research.

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

## APPENDIX

You may include other additional sections here.

## A1 CASE-CONTROL SAMPLE BREAKDOWN WITH $k$ TIME POINTS AND CENSORING

Table A1: Case-control sample breakdown with $k$ time points and censoring

| | size | weight | group | count |
|---|---|---|---|---|
| time 0 censored | $K_0 c_0$ | $\ell_0$ | $G_0$ | $\ell_0 \cdot \frac{K_0 c_0}{s_0}$ |
| time 0 case | $c_0$ | $w_0$ | $G_1$ | $w_0 \cdot 1$ |
| time 0 control | $J_0 c_0$ | $m_0$ | $G_2, \ldots, G_{2k}$ | $m_0 \cdot \frac{J_0 c_0}{N - c_0 - s_0}$ |
| time 1 censored | $K_1 c_1$ | $\ell_1$ | $G_2$ | $\ell_1 \cdot \frac{K_1 c_1}{s_1}$ |
| time 1 case | $c_1$ | $w_1$ | $G_3$ | $w_1 \cdot 1$ |
| time 1 control | $J_1 c_1$ | $m_1$ | $G_4, \ldots, G_{2k}$ | $m_1 \cdot \frac{J_1 c_1}{N - c_0 - s_0 - c_1 - s_1}$ |
| $\vdots$ | $\vdots$ | $\vdots$ | $\vdots$ | $\vdots$ |
| time $k-1$ censored | $K_{k-1} c_{k-1}$ | $\ell_{k-1}$ | $G_{2k-2}$ | $\ell_{k-1} \cdot \frac{K_{k-1} c_{k-1}}{s_{k-1}}$ |
| time $k-1$ case | $c_{k-1}$ | $w_{k-1}$ | $G_{2k-1}$ | $w_{k-1} \cdot 1$ |
| time $k-1$ control | $J_{k-1} c_{k-1}$ | $m_{k-1}$ | $G_{2k}$ | $m_{k-1} \cdot \frac{J_{k-1} c_{k-1}}{N - \sum_{j=0}^{k-1} c_j - \sum_{j=0}^{k-1} s_j}$ |

For the case-control sample to be representative of the full sample, we require that each individual in groups $G_0$ to $G_{2k}$ be counted the same number of times. Therefore, we may set

$$\ell_0 \frac{K_0 c_0}{s_0} = w_0 = m_0 \frac{J_0 c_0}{N - c_0 - s_0} + \ell_1 \frac{K_1 c_1}{s_1}$$

$$= m_0 \frac{J_0 c_0}{N - c_0 - s_0} + w_1 = m_0 \frac{J_0 c_0}{N - c_0 - s_0} + m_1 \frac{J_1 c_1}{N - c_0 - s_0 - c_1 - s_1} + \ell_2 \frac{K_2 c_2}{s_2}$$

$$\vdots$$

$$= \sum_{i=0}^{k-2} m_i \frac{J_i c_i}{N - \sum_{j=0}^{i} c_j - \sum_{j=0}^{i} s_j} + \ell_{k-1} \frac{K_{k-1} c_{k-1}}{s_{k-1}}$$

$$= \sum_{i=0}^{k-2} m_i \frac{J_i c_i}{N - \sum_{j=0}^{i} c_j - \sum_{j=0}^{i} s_j} + w_{k-1} = \sum_{i=0}^{k-1} m_i \frac{J_i c_i}{N - \sum_{j=0}^{i} c_j - \sum_{j=0}^{i} s_j},$$

which reduces to (4) in the main context.

## A2 Algorithm of ICE Estimator

The algorithm for deriving the unweighted ICE estimator is described in detail below.

---

**Algorithm A1** ICE Estimator for Estimating $E(Y_T^g)$

---

1: Regress $Y_T$ on $\bar{L}_{T-1}$ and $\bar{A}_{T-1}$ among individuals with $Y_{T-1} = C_T = 0$, and estimate parameter $\theta_{T,T-1}$.

2: Obtain predicted values $\hat{h}_{T,T-1}^g$ from $E(Y_T \mid Y_{T-1} = C_T = 0, \bar{L}_{T-1}, \bar{A}_{T-1} = \bar{A}_{T-1}^g; \hat{\theta}_{T,T-1})$ by fixing $\bar{A}_{T-1} = \bar{A}_{T-1}^g$ for all individuals with $Y_{T-1} = C_{T-1} = 0$. Set $q = 2$.

3: **while** $q \leq T$ **do**

4:     Let $k = T - q$.

5:     Define

$$\hat{Q}_{T,k+1}^g = \begin{cases} \hat{h}_{T,k+1}^g & \text{if } Y_{k+1} = 0, \\ 1 & \text{if } Y_{k+1} = 1. \end{cases}$$

6:     Regress $\hat{Q}_{T,k+1}^g$ on $\bar{L}_k$ and $\bar{A}_k$ among individuals with $Y_k = C_{k+1} = 0$, and estimate parameter $\theta_{T,k}$.

7:     Obtain predicted values $\hat{h}_{T,k}^g$ from $E(\hat{Q}_{T,k+1}^g \mid Y_k = C_{k+1} = 0, \bar{L}_k, \bar{A}_k = \bar{A}_k^g; \hat{\theta}_{T,k})$ by fixing $\bar{A}_k = \bar{A}_k^g$ among individuals with $Y_k = C_k = 0$.

8:     Set $q = q + 1$.

9: **end while**

10: Average $\hat{h}_{T,0}^g$ over all individuals to estimate $E(Y_T^g)$.

---

## A3 Additional Weight Combinations

Below we give 4 sets of possible weights for the illustrative example with 2 time points without censoring:

(1) Let $w_0 = w_1 = 1$, we have

$$S(O) = c_0 w_0 S(O_{0c}) + c_1 w_0 S(O_{1c}) + (N - c_0 - c_1) w_0 S(O_{1m}),$$

and $m_0 = 0, m_1 = \frac{N - c_0 - c_1}{J c_1}$.

(2) Let $w_1 = m_1 = 0$, we have

$$S(O) = c_0 w_0 S(O_{0c}) + (N - c_0) w_0 S(O_{0m}),$$

and $w_0 = 1, m_0 = \frac{N - c_0}{J c_0}$.

(3) Let $m_0 = m_1 = 1/J$, we have

$$S(O) = c_1 w_1 S(O_{1c}) + c_1 S(O_{1m}) + c_0 w_0 S(O_{0c}) + c_0 S(O_{0m}),$$

where $w_1 = \frac{c_1}{N - c_0 - c_1}$, $w_0 = \frac{c_1}{N - c_0 - c_1} + \frac{c_0}{N - c_0}$.

(4) Let $m_0 = \frac{N - c_0}{JN}$, $m_1 = \frac{N - c_0 - c_1}{J(N - c_0)}$, then $w_0 = \frac{c_0}{N} + \frac{c_1}{N - c_0}$, $w_1 = \frac{c_1}{N - c_0}$,

$$S(O) = c_1 w_1 S(O_{1c}) + J c_1 m_1 S(O_{1m}) + c_0 w_0 S(O_{0c}) + J c_0 m_0 S(O_{0m}).$$

## A4    PROOF OF EQUATION (3)

*Proof.* Let

$$S(O) = c_1 w_1 S(O_{1c}) + J c_1 m_1 S(O_{1m}) + c_0 w_0 S(O_{0c}) + J c_0 m_0 S(O_{0m})$$
$$= c_1 w_1 S(O_{1c}) + (N - c_0 - c_1) w_1 S(O_{1m}) + c_0 w_0 S(O_{0c}) + (w_0 - w_1)(N - c_0) S(O_{0m}),$$

then

$E\{S(O)\}$

$$= c_1 w_1 \int S_0(O^*) f(O^* \mid \text{die at } t = 1) dO^* + J c_1 m_1 \int S_0(O^*) f(O^* \mid \text{alive at } t = 1) dO^*$$

$$+ c_0 w_0 \int S_0(O^*) f(O^* \mid \text{die at } t = 0) dO^* + J c_0 m_0 \int S_0(O^*) f(O^* \mid \text{alive at } t = 0) dO^*$$

$$= \frac{c_1 w_1}{\Pr(\text{die at } t = 1)} \int S_0(O^*) f(O^*, \text{die at } t = 1) dO^* + \frac{J c_1 m_1}{\Pr(\text{alive at } t = 1)} \int S_0(O^*) f(O^*, \text{alive at } t = 1) dO^*$$

$$+ \frac{c_0 w_0}{\Pr(\text{die at } t = 0)} \int S_0(O^*) f(O^*, \text{die at } t = 0) dO^* + \frac{J c_0 m_0}{\Pr(\text{alive at } t = 0)} \int S_0(O^*) f(O^*, \text{alive at } t = 0) dO^*$$

$$= c_1 w_1 \frac{N}{c_1} \int S_0(O^*) f(O^*, \text{die at } t = 1) dO^* + J c_1 m_1 \frac{N}{N - c_0 - c_1} \int S_0(O^*) f(O^*, \text{alive at } t = 1) dO^*$$

$$+ c_0 w_0 \frac{N}{c_0} \int S_0(O^*) f(O^*, \text{die at } t = 0) dO^* + J c_0 m_0 \frac{N}{N - c_0} \int S_0(O^*) f(O^*, \text{alive at } t = 0) dO^*$$

$$= w_1 N \int S_0(O^*) f(O^*, \text{alive at } t = 0) dO^* + c_0 w_0 \frac{N}{c_0} \int S_0(O^*) f(O^*, \text{die at } t = 0) dO^*$$

$$+ J c_0 m_0 \frac{N}{N - c_0} \int S_0(O^*) f(O^*, \text{alive at } t = 0) dO^*$$

$$= w_0 N \int S_0(O^*) f(O^*) dO^*$$

$$= w_0 N E_0^*\{S_0(O^*)\}.$$

If we let $w_0 = N^{-1}$, then $E\{S(O)\} = E_0^*\{S_0(O^*)\}$. But if we only need $E\{S(O)\} = w_0 N E_0^*\{S_0(O^*)\} = 0$, then it is not necessary to set $w_0 = N^{-1}$. □

## A5    PROOF OF THEOREM 1

We have already known that the ICE estimator in Algorithm A1 is consistent given that the models are correctly specified. To show that the estimator in Algorithm 1 to be consistent, we only need to prove that at each step, the expectation of the new estimating equation based on the case-control sample is zero if and only if the expectation of the original estimating equation is zero.

*Proof.* Let

$$S(O) = c_{k-1} w_{k-1} S(O_{k-1,c}) + K_{k-1} c_{k-1} \ell_{k-1} S(O_{k-1,s}) + J_{k-1} c_{k-1} m_{k-1} S(O_{k-1,m})$$
$$+ \cdots + c_0 w_0 S(O_{0c}) + K_0 c_0 \ell_0 S(O_{0s}) + J_0 c_0 m_0 S(O_{0m}),$$

then

$E\{S(O)\}$

$$= c_{k-1} w_{k-1} \int S_0(O^*) f(O^* \mid \text{die at } t = k - 1) dO^* + K_{k-1} c_{k-1} \ell_{k-1} \int S_0(O^*) f(O^* \mid \text{censored at } t = k - 1) dO^*$$

$$+ J_{k-1} c_{k-1} m_{k-1} \int S_0(O^*) f(O^* \mid \text{alive at } t = k - 1) dO^* + \cdots$$

$$+ c_0 w_0 \int S_0(O^*) f(O^* \mid \text{die at } t = 0) dO^* + K_0 c_0 \ell_0 \int S_0(O^*) f(O^* \mid \text{censored at } t = 0) dO^*$$

$$+ J_0 c_0 m_0 \int S_0(O^*) f(O^* \mid \text{alive at } t = 0) dO^*$$

$$= \frac{c_{k-1} w_{k-1}}{\Pr(\text{die at } t = k-1)} \int S_0(O^*) f(O^*, \text{die at } t = k-1) dO^* + \frac{K_{k-1} c_{k-1} \ell_{k-1}}{\Pr(\text{censored at } t = k-1)} \int S_0(O^*) f(O^*, \text{censored at } t = k-1) dO^*$$

$$+ \frac{J_{k-1} c_{k-1} m_{k-1}}{\Pr(\text{alive at } t = k-1)} \int S_0(O^*) f(O^*, \text{alive at } t = k-1) dO^* + \cdots$$

$$+ \frac{c_0 w_0}{\Pr(\text{die at } t = 0)} \int S_0(O^*) f(O^*, \text{die at } t = 0) dO^* \frac{K_0 c_0 \ell_0}{\Pr(\text{censored at } t = 0)} \int S_0(O^*) f(O^*, \text{censored at } t = 0) dO^*$$

$$+ \frac{J c_0 m_0}{\Pr(\text{alive at } t = 0)} \int S_0(O^*) f(O^*, \text{alive at } t = 0) dO^*$$

$$= c_{k-1} w_{k-1} \frac{N}{c_{k-1}} \int S_0(O^*) f(O^*, \text{die at } t = k-1) dO^* + K_{k-1} c_{k-1} \ell_{k-1} \frac{N}{s_{k-1}} \int S_0(O^*) f(O^*, \text{censored at } t = k-1) dO^*$$

$$+ J_{k-1} c_{k-1} m_{k-1} \frac{N}{N - \sum_{j=0}^{k-1} c_j - \sum_{j=0}^{k-1} s_j} \int S_0(O^*) f(O^*, \text{alive at } t = k-1) dO^* + \cdots$$

$$+ c_0 w_0 \frac{N}{c_0} \int S_0(O^*) f(O^*, \text{die at } t = 0) dO^* + K_0 c_0 \ell_0 \frac{N}{s_0} \int S_0(O^*) f(O^*, \text{censored at } t = 0) dO^*$$

$$+ J_0 c_0 m_0 \frac{N}{N - c_0 - s_0} \int S_0(O^*) f(O^*, \text{alive at } t = 0) dO^*$$

$$= w_{k-1} N \int S_0(O^*) f(O^*, \text{die at } t = k-1) dO^* + w_{k-1} N \int S_0(O^*) f(O^*, \text{censored at } t = k-1) dO^*$$

$$+ w_{k-1} N \int S_0(O^*) f(O^*, \text{alive at } t = k-1) dO^* + \cdots$$

$$+ w_0 N \int S_0(O^*) f(O^*, \text{die at } t = 0) dO^* + w_0 N \int S_0(O^*) f(O^*, \text{censored at } t = 0) dO^*$$

$$+ J c_0 m_0 \frac{N}{N - c_0} \int S_0(O^*) f(O^*, \text{alive at } t = 0) dO^*$$

$$= w_{k-1} N \int S_0(O^*) f(O^*, \text{alive at } t = k-2) dO^* + w_{k-2} N \int S_0(O^*) f(O^*, \text{die at } t = k-2) dO^*$$

$$+ w_{k-2} N \int S_0(O^*) f(O^*, \text{censored at } t = k-2) dO^* + (w_{k-2} - w_{k-1}) N \int S_0(O^*) f(O^*, \text{alive at } t = k-2) dO^*$$

$$+ \cdots + w_0 N \int S_0(O^*) f(O^*, \text{die at } t = 0) dO^* + w_0 N \int S_0(O^*) f(O^*, \text{censored at } t = 0) dO^*$$

$$+ (w_0 - w_1) N \int S_0(O^*) f(O^*, \text{alive at } t = 0) dO^*$$

$$= w_0 N \int S_0(O^*) f(O^*) dO^*$$

$$= w_0 N E_0^* \{ S_0(O^*) \}$$

Therefore, $E\{S(O)\} = 0$ if and only if $E_0^* \{ S_0(O^*) \} = 0$. □

## A6 MORE NUMERICAL EXAMPLES

We also tried two more simulation setups, with only the model of $Y$ changing to

$$Y_t \sim \text{Ber}\big(\text{expit}(-4 + 2A_t + \beta^\top L_{t-1})\big),$$

and

$$Y_t \sim \text{Ber}\big(\text{expit}(-5 + 2A_t + \beta^\top L_{t-1})\big)$$

corresponding to different event rates, referred to medium/high prevalence scenario. The results are shown in Tables A3-A4, and Figures A1-A2.

Table A2: Summary of the computation time (in mins) with superlearner.

|  | ICE ($\bar{A} = \bar{0}$) | ICE ($\bar{A} = \bar{1}$) |
|---|---|---|
| Complete | 63.32 (6.19) | 62.97 (6.32) |
| (*model-fitting time*) | | |
| Case-control (J=20) | 5.42 (0.75) | 5.58 (0.79) |
| Case-control (J=10) | 3.12 (0.51) | 3.21 (0.52) |
| (*total runtime*) | | |
| Case-control (J=20) | 5.44 (0.76) | 5.60 (0.80) |
| Case-control (J=10) | 3.14 (0.52) | 3.23 (0.53) |

Table A3: Summary of the computation time for medium prevalence scenario (in secs)

|  | NICE, $\bar{A} = \bar{0}$ | ICE, $\bar{A} = \bar{0}$ | NICE, $\bar{A} = \bar{1}$ | ICE, $\bar{A} = \bar{1}$ |
|---|---|---|---|---|
| Complete | 11.26 (1.80) | 4.22 (1.03) | 11.28 (1.97) | 3.91 (1.00) |
| (*model-fitting time*) | | | | |
| Case-control (J=20) | 8.27 (2.23) | 3.02 (0.90) | 8.30 (2.32) | 2.85 (0.89) |
| Case-control (J=10) | 6.92 (1.96) | 2.04 (0.86) | 6.89 (1.97) | 1.95 (1.01) |
| Case-control (J=5) | 6.16 (1.59) | 1.40 (0.47) | 6.15 (1.33) | 1.35 (0.47) |
| (*total runtime*) | | | | |
| Case-control (J=20) | 9.92 (2.78) | 3.89 (1.26) | 9.91 (2.75) | 3.74 (1.28) |
| Case-control (J=10) | 8.48 (2.62) | 2.94 (1.53) | 8.50 (2.93) | 2.86 (1.75) |
| Case-control (J=5) | 7.35 (2.08) | 2.04 (0.91) | 7.33 (1.79) | 1.98 (0.86) |

Table A4: Summary of the computation time for high prevalence scenario (in secs)

|  | NICE, $\bar{A} = \bar{0}$ | ICE, $\bar{A} = \bar{0}$ | NICE, $\bar{A} = \bar{1}$ | ICE, $\bar{A} = \bar{1}$ |
|---|---|---|---|---|
| Complete | 11.06 (2.33) | 4.44 (2.44) | 11.22 (2.60) | 4.01 (2.26) |
| (*model-fitting time*) | | | | |
| Case-control (J=20) | 12.38 (2.35) | 5.21 (1.94) | 12.41 (2.54) | 4.79 (1.88) |
| Case-control (J=10) | 8.30 (1.99) | 3.26 (1.63) | 8.27 (1.83) | 3.02 (1.59) |
| Case-control (J=5) | 7.31 (1.76) | 2.44 (1.30) | 7.30 (1.70) | 2.33 (1.40) |
| (*total runtime*) | | | | |
| Case-control (J=20) | 14.20 (2.94) | 6.49 (2.72) | 14.25 (3.09) | 6.08 (2.64) |
| Case-control (J=10) | 9.96 (2.60) | 4.31 (2.27) | 9.93 (2.48) | 4.06 (2.23) |
| Case-control (J=5) | 8.94 (2.42) | 3.36 (1.98) | 8.91 (2.28) | 3.25 (2.24) |

Remark: Since the prevalence is approximately 5–8%, when $J = 20$ the sample size exceeded the original sample sizes, and consequently the time also exceeded that of the complete-data version.

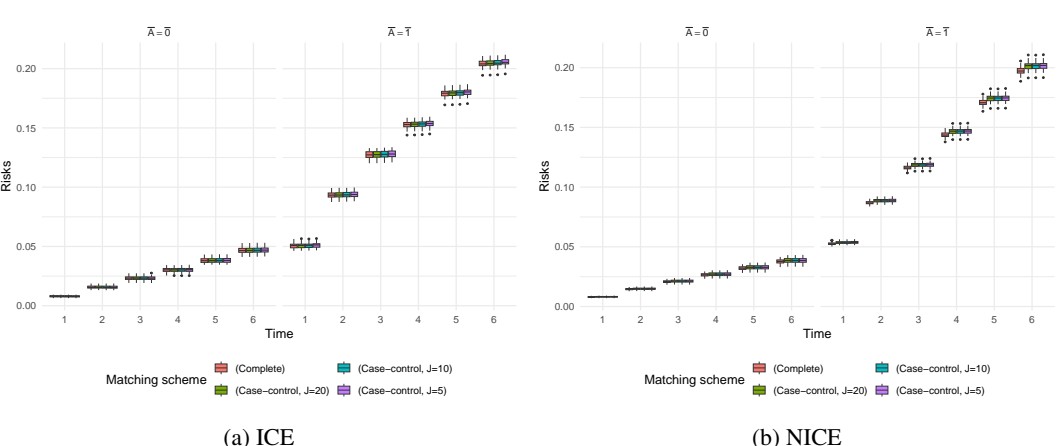

(a) ICE

(b) NICE

Figure A1: Risk estimates for always or never exposed to treatment for medium prevalence scenario.

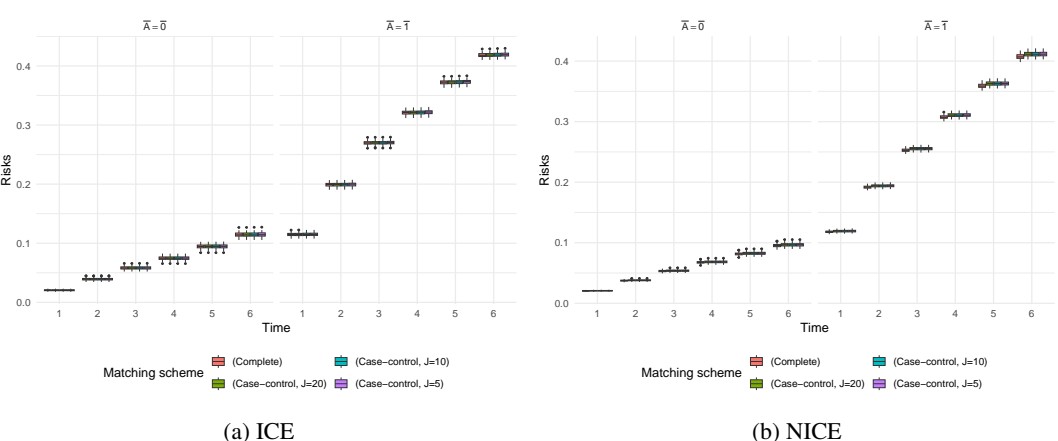

(a) ICE

(b) NICE

Figure A2: Risk estimates for always or never exposed to treatment for high prevalence scenario.

## A7   SBDH, CLINICAL COMORBIDITIES, AND MENTAL HEALTH DISORDERS

**Structured SBDH categories**. We used ICD-10-CM Z codes [2] to extract information about 10 structured SDOH categories -

1. Problems related to education and literacy (**Z55**),
2. Problems related to employment and unemployment (**Z56**),
3. Occupational exposure to risk factors (**Z57**),
4. Problems related to housing and economic circumstances (**Z59**),
5. Problems related to social environment (**Z60**),
6. Problems related to upbringing (**Z62**),
7. Other problems related to primary support group, including family circumstances (**Z63**),
8. Problems related to certain psychosocial circumstances (**Z64**),
9. Problems related to other psychosocial circumstances (**Z65**), and
10. Problems related to medical facilities and other health care (**Z75**)

**NLP-extracted SBDH categories**. We considered 12 NLP-extracted SBDH categories - Social isolation, job or financial insecurity, housing instability, legal problems, violence, barriers to care, transition of care, food insecurity, psychiatric symptoms, substance abuse, pain, and patient disability. Definitions and sample examples of all SBDH categories as well as the deep-learning model used to extract these SBDH information are available in the appendix section of the study by Mitra et al. (Mitra et al., 2023).

**Clinical comorbidities and mental health disorders**. From the Charlson Comorbidity Index (Quan et al., 2011), we included 17 clinical comorbidities: acute myocardial infarction, congestive heart failure, peripheral vascular disease, cerebrovascular disease, dementia, chronic obstructive pulmonary disease, rheumatoid disease, peptic ulcer disease, mild liver disease, diabetes without complications, diabetes with complications, hemiplegia or paraplegia, kidney disease, cancer, moderate or severe liver disease, metastatic solid tumor, and AIDS/HIV. We considered 7 mental health disorders (Blosnich et al., 2020): major depressive disorder, alcohol use disorder, drug use disorder, anxiety disorder, posttraumatic stress disorder, schizophrenia, and bipolar disorder.

Table A5: Summary of the real dataset.

| Time | Total # | # of Cases | # of Controls | # of Censored | Censor Rate | Event Rate |
|------|---------|------------|---------------|---------------|-------------|------------|
| 1 | 125,928 | 331 | 124,351 | 1,473 | 1.17% | 0.27% |
| 2 | 124,351 | 227 | 122,673 | 1,591 | 1.28% | 0.18% |
| 3 | 122,673 | 76 | 102,047 | 20,550 | 16.755% | 0,07% |
| 4 | 102,047 | 64 | 76,537 | 25,466 | 24.96% | 0.08% |
| 5 | 76,537 | 20 | 48,008 | 28,509 | 37.25% | 0.04% |

---

[2]https://www.bcbsnm.com/docs/provider/nm/icd-10-z-codes.pdf

Table A6: Summary Statistics of the real dataset.

| Variables | | Case (n=440), % | Control (n=126,959), % |
|---|---|---|---|
| Race | White | 369 (83.86) | 81,443 (64.15) |
| | Black | 43 (9.77) | 34,536 (27.20) |
| | Others | 28 (6.36) | 10,980 (8.65) |
| Sex | Male | 403 (91.59) | 113,690 (89.55) |
| | Female | 37 (8.41) | 13,269 (10.45) |
| Age | 18-29 | 74 (16.82) | 11,956 (9.42) |
| | 30-39 | 97 (22.05) | 23,731 (18.69) |
| | 40-49 | 78 (17.73) | 18,918 (14.90) |
| | 50-59 | 94 (21.36) | 34,597 (27.25) |
| | 60-69 | 73 (16.59) | 29,067 (22.89) |
| | 70-79 | 22 (5.00) | 6,625 (5.22) |
| | 79< | 2 (0.45) | 2,065 (1.63) |
| Marital Status | Married | 71 (16.14) | 11,722 (9.23) |
| | Not married | 163 (37.05) | 25,031 (19.72) |
| | Unknown | 206 (46.82) | 90,206 (71.05) |

## A8 GUIDANCE ON SELECTING WEIGHTING SCHEMES / SAMPLING RATIO

Our framework allows a family of valid weighting schemes that lead to statistically consistent estimation, provided that the weights satisfy the balance equations in (4). While different choices of weights and sampling ratios do not affect consistency, they indeed may influence efficiency. Unfortunately, deriving closed-form variance formulas for either the NICE or the ICE implementation of the g-formula is prohibitively complex, rendering analytic efficiency comparisons across weighting choices difficult to carry out in practice. Accordingly, the prevailing practice in applied g-formula work is to obtain standard errors via non-parametric bootstrap resampling (Young et al., 2011; Keil et al., 2014; Wen et al., 2021), and we follow the same practice throughout our simulations and real-data application.

As mentioned above, the variance should not be sensitive to the case-control ratio as long as the number of controls becomes adequate (Wang, 2020). We recommend a pragmatic procedure for selecting the sampling ratios as follows.

(i) begin with five controls per case; this ratio is commonly used in EHR applications.

(ii) if events become sparse in later periods, raise adaptively until each regression retains a reasonable effective sample size (e.g., observations); and

(iii) run a short pilot bootstrap comparing two or three candidate ratios to gauge the marginal drop in standard error versus added computation time.

In practice, this empirical tuning typically finds a ratio that preserves nearly all of the precision of the full-data estimator while still delivering substantial speed-ups.

## A9 NOTATION TABLE

For clarity, we provide in Table A7 a summary of the notation used used throughout the paper.

Table A7: Summary of the key notation used throughout the paper.

| Symbol | Definition |
|---|---|
| $N$ | Total number of individuals in the study population |
| $T$ | Maximum number of discrete follow-up time points |
| $L_j$ | Covariates measured at time $j$ |
| $A_j$ | Treatment indicator at time $j$ |
| $C_{j+1}$ | Censoring indicator at time $j+1$ (1 = censored, 0 = observed) |
| $Y_{j+1}$ | Event indicator at time $j+1$ (1 = event occurred, 0 = no event) |
| $\bar{L}_j$ | History of covariates up to time $j$ |
| $\bar{A}_j$ | History of treatments up to time $j$ |
| $Y_t^g$ | Potential survival outcome at time $t$ under strategy $g$ |
| $E(Y_t^g)$ | Counterfactual risk under strategy $g$ at time $t$ |
| $g$ | Deterministic treatment strategy mapping histories to assigned treatments |
| $c_t$ | Number of cases (events) occurring at time $t$ |
| $s_t$ | Number of censored individuals at time $t$ |
| $J_t$ | Sampling ratio for controls (number of controls per case) at time $t$ |
| $K_t$ | Sampling ratio for censored individuals (number of censored individuals per case) at time $t$ |
| $w_t$ | Sampling weight assigned to cases at time $t$ |
| $m_t$ | Sampling weight assigned to controls at time $t$ |
| $\ell_t$ | Sampling weight assigned to censored individuals at time $t$ |
| $\hat{h}_{T,k}^g$ | Predicted conditional expectation at time $k$ under strategy $g$ |
| $\hat{Q}_{T,k}^g$ | Recursively defined pseudo-outcome used in ICE estimation |
| $\theta_{T,k}$ | Regression parameters estimated at time $k$ |

