# OpenReview forum: "A Computationally Efficient Case-Control Sampling Framework for G-Formula with Longitudinal Data"
_ICLR.cc/2026/Conference — ICLR 2026 Conference Withdrawn Submission_

### Official Review · Reviewer_TBcK · 2025-10-15

**Soundness:** 2
**Presentation:** 2
**Contribution:** 1
**Rating:** 2
**Confidence:** 4

**Summary:**

The paper proposes a subsampling method for increasing the computational efficiency of G-computation for longitudinal causal inference.
The method relies on subsampling and upweighting the control group at different time steps to reduce the effective sample size while still enabling unbiased estimation.
The method is evaluated using both synthetic and real-world longitudinal data.

**Strengths:**

- The method increases computational efficiency and retains unbiasedness of the G-formula estimator

- The method is applicable in various application domains such as medicine

**Weaknesses:**

- I suspect that subsampling comes **at the cost of increasing estimation variance**, which the authors did not discuss in their paper.
Subsampling implies that we essentially decrease our sample size. I believe it is crucial to analyze this and provide guidelines on how to choose.

- The novelty is limited. Both G-computation and subsampling are **well-established methods** which the authors combine in their work.

- I do not think that Eq. (1) and (2) are the correct identification formulas under censoring. For censoring, identification usually either follows via hazard functions or inverse censoring weights.

- The empirical improvements are mostly in the 5-10 second range and do not seem practically relevant.

- The writing and general presentation of the paper could be improved.

**Questions:**

- Can the authors double check the correctness of the G-formula under censoring?

- How does the proposed method perform in more computationally demanding settings (e.g., when using ML models instead of logistic regressions)? The empirical improvements are mostly in the 5-10 second range and do not seem practically relevant.

---

### Official Review · Reviewer_NEeR · 2025-10-30

**Soundness:** 3
**Presentation:** 3
**Contribution:** 2
**Rating:** 4
**Confidence:** 3

**Summary:**

To address the computational burden of using bootstrapping for variance estimation with iterative conditional expectation, particularly in the context of rare outcomes, this paper proposes a case-control enhanced g-formula.

The proposed method has two key components. The first is the g-formula, which is commonly used in longitudinal settings to alleviate weight instability in causal inference. The second component is case-control sub-sampling. This strategy is analogous to up-sampling (or retaining) the "case" subjects and matching them with a sufficiently large set of "control" subjects with weighting. This enables more efficient parameter estimation while significantly reducing computational cost.

**Strengths:**

I appreciate the connection the authors make between the sub-sampling strategy and the weighted estimating equation, which allows for the straightforward incorporation of censoring in survival outcomes. The resulting estimator is shown to be consistent under reasonable assumptions.

The numerical experiments align with theoretical expectations:

* The case-control enhanced g-formula is shown to be less time-consuming, particularly when a complex ensemble model like Super Learner is used (as shown in Table A2).
* The estimator produced by the case-control sampling method yields results that are similar to those from the complete-data scenario. It is also noted that higher sampling ratios lead to estimated standard errors that more closely approximate the complete-data scenario.

In addition, it is promising to see that the case-control estimator achieves results comparable to the complete-data approach while requiring significantly less computation time.

**Weaknesses:**

* Since the case-control sub-sampling strategy may also address the weight instability by drawing informative subsets of the control data, why combining g-formula with the case-control sampling strategy, instead of MSM or SNM with the case-control sampling? Since the g-formula requires the correct model specification of all sub-models, it seems quite restrictive compared to MSM/SNM which offers the double robustness (I.e., correct either treatment or outcome model) and semi-parametric efficiency.

* Based on the box plots of risks in Figure 1, the case-control estimator seems to have similar empirical standard errors as the complete-data scenario. As the estimated standard errors are larger for smaller sampling ratios, I would expect some over-coverage issue in these cases, which might affect the power of the estimator. I suspect the case-control sampling strategy can be used in conjunction with any other estimating equation (besides the g-formula). Did author consider some more efficient and doubly robust estimation equations as candidates?

**Questions:**

* What is the coverage of the estimator across the 100 simulated datasets? The results only show the estimated standard error from bootstrapping.

* What models were used for the real-data application? My guess would be a SuperLearner, but the running time of approximately 25 secs reported in Table 4 seems very fast for such a model.

---

### Official Review · Reviewer_xhkF · 2025-10-31

**Soundness:** 3
**Presentation:** 1
**Contribution:** 1
**Rating:** 2
**Confidence:** 3

**Summary:**

The authors consider the problem of estimating a time-varying treatment effect on survival data with rare events. The desired quantity can be estimated using the "g-formula" via two existing methods, ICE and NICE. They identify two problems with these existing methods. Namely, when the methods are fit using a large dataset, the computational cost is high, especially since there are no closed-form expressions for the estimator variance this variance is generally approximated via bootstrap sampling, further exacerbating the computational cost. Second, when events are rare, there is a large class imbalance between failure/non-failure at each time point, which can lead to slow convergence for model fitting. They propose a subsampling/importance reweighting approach to address both of these issues. The subsampling to a smaller dataset reduces the computational cost, while the importance reweighting means that a comparable number of failure/non-failure datapoints can be sampled without introducing bias. They confirm that their approach gives similar output to the original method fit to the whole dataset on both real and synthetic data.

**Strengths:**

The subsampling approach introduced by the authors does in fact obtain a large speedup over fitting to the whole dataset. Based on their experiments, it seems that the output of their subsampled method is consistent with the full dataset method.

**Weaknesses:**

I found the paper difficult to follow. Some formulas are not clearly stated, or quantities not explicitly defined. For instance, one can infer the definition of $f\_{\bar{L}\_j, \bar{A}\_{j-1}, C\_j, Y\_j}(\bar{l}\_j, \bar{a}^g\_{j-1},0,0)$ from context, but especially given the intricacy of the formulas it would help to state it explicitly. Another instance which impedes understanding of the paper/results is equation (2). I am not actually sure how to fill in the ... in the formula. Perhaps these quantities are clear to a reader who is familiar with the g-formula, but it was not obvious to me.

Adding to this difficulty is the fact that some critical pieces of the paper are relegated to appendices. For example, the definition of the main comparison algorithm (Algorithm A1) is in an appendix, though its description is referred to extensively. For the main runtime reduction results on the synthetic data, one must compare the results of the proposed algorithm (presented in seconds) in Table 2 with the results of the baseline algorithm (presented in minutes) in appendix Table A2.

The example starting on line 161 is also presented in a confusing manner. To begin with, it is not stated why we want to estimate the expected value of a covariate X, nor what the relationship is between X and the quantitites L, C, A, and Y introduced in the problem setup. The presentation of the weighting scheme could also use improvement, as it consists of several dense paragraphs with equations interspersed both inline and in standalone equations. More importantly, the final goal of the choice of weights, or even the point of the section, is not clear with respect to the main goal of the paper, i.e., to estimate the time-varying treatment effect. It is only at the *end* of the section that it is explained that the weighting scheme is just debiasing an expected gradient calculation used to fit the models at each time step (and the explanation of this also uses quite non-standard notation).

Table 3 presents the bootstap means and standard error of the ICE and NICE estimates, but it is not clear how to interpret these results. Since we have the ground truth for the synthetic data, it would make more sense to present the error of each method. Or, if the main point is just that the proposed subsampling approach is a good approximation of the more computationally intensive complete-data method, then the error of these estimators to the full-data estimate should be provided.

In addition to the general problems with clarity, the novelty of the contribution itself is limited. After digging past the notation, the proposed method boils down to a standard class-imbalanced importance reweighting technique to subsample a large dataset without introducing bias to gradient calculations. The computational speedups obtained are purely from the smaller dataset size, and the ICE and NICE methods can be applied directly (with the standard importance weights) on the smaller dataset.

Typos:
1. The appendix starts with the filler text "You may include additional sections here."

**Questions:**

Does the consistency assumption mean that two individuals with the same treatment history necessarily have the same covariate sequence?  If so, doesn't this make the covariates $L_j$ redundant, as they can be recovered from the treatment sequence?

---

### Note · Authors · 2025-11-21

I have read and agree with the venue's withdrawal policy on behalf of myself and my co-authors.